# Complex interventions for aggressive challenging behaviour in adults with intellectual disability: A rapid realist review informed by multiple populations

Rachel Royston[1]*, Stephen Naughton[1], Angela Hassiotis[1], Andrew Jahoda[2], Afia Ali[1¤], Umesh Chauhan[3], Sally-Ann Cooper[2], Athanasia Kouroupa[1], Liz Steed[4], Andre Strydom[5], Laurence Taggart[6], Penny Rapaport[1]

1 Division of Psychiatry, University College London, London, United Kingdom, 2 School of Health & Wellbeing, University of Glasgow, Glasgow, United Kingdom, 3 School of Medicine, University of Central Lancashire, Lancashire, United Kingdom, 4 Wolfson Institute of Population Health, Queen Mary, University of London, London, United Kingdom, 5 Forensic & Neurodevelopmental Sciences, King's College London, London, United Kingdom, 6 School of Nursing and Paramedic Science, Ulster University, Northern Ireland, United Kingdom

¤ Current address: Unit for Social and Community Psychiatry, Queen Mary University of London, London, United Kingdom
* r.royston@ucl.ac.uk

**Data Availability Statement:** All relevant data are within the paper and its Supporting Information files.

## Abstract

### Objectives

Approximately 10% of people with intellectual disability display aggressive challenging behaviour, usually due to unmet needs. There are a variety of interventions available, yet a scarcity of understanding about what mechanisms contribute to successful interventions. We explored how complex interventions for aggressive challenging behaviour work in practice and what works for whom by developing programme theories through contexts-mechanism-outcome configurations.

### Methods

This review followed modified rapid realist review methodology and RAMESES-II standards. Eligible papers reported on a range of population groups (intellectual disability, mental health, dementia, young people and adults) and settings (community and inpatient) to broaden the scope and available data for review.

### Results

Five databases and grey literature were searched and a total of 59 studies were included. We developed three overarching domains comprising of 11 contexts-mechanism-outcome configurations; 1. Working with the person displaying aggressive challenging behaviour, 2. Relationships and team focused approaches and 3. Sustaining and embedding facilitating factors at team and systems levels. Mechanisms underlying the successful application of

**Funding:** This paper presents independent research commissioned and funded by the National Institute for Health Research (NIHR) Programme Grants for Applied Research (grant number; NIHR200120). This award was received by AH, AJ, AA, UC, SC, LS, AS, LT & PR. The views expressed are those of the authors and not necessarily those of the NIHR or the Department of Health and Social Care. The funders had no role in study design, data collection and analysis, decision to publish, or preparation of the manuscript.

**Competing interests:** The authors have declared that no competing interests exist.

**Abbreviations:** CBT, Cognitive Behavioural Therapy; CMO, Context-Mechanisms-Outcomes; LRG, Local Reference Group; NIHR, National Institute for Health Research.

interventions included improving understanding, addressing unmet need, developing positive skills, enhancing carer compassion, and boosting staff self-efficacy and motivation.

## Conclusion

The review emphasises how interventions for aggressive challenging behaviour should be personalised and tailored to suit individual needs. Effective communication and trusting relationships between service users, carers, professionals, and within staff teams is essential to facilitate effective intervention delivery. Carer inclusion and service level buy-in supports the attainment of desired outcomes. Implications for policy, clinical practice and future directions are discussed.

## Prospero registration number

CRD42020203055.

## Introduction

Aggressive challenging behaviour is defined as any non-verbal, verbal or physical behaviour perceived to be threatening or that causes harm to others or property [1, 2]. The display of clinically significant aggressive challenging behaviour in adults with intellectual disability is common, occurring as often as weekly in 7–10% of adults [3]. Higher rates have also been reported, varying from 31–75% depending on the type of aggressive challenging behaviour and population in question [4, 5]. There are a range of triggers of aggressive challenging behaviour for people with intellectual disability, including placing demands on the person, provocation from others, changing activities or unexpected events [2]. These behaviours are a primary driver for the use of restrictive practices and psychiatric admission of people with intellectual disability in the absence of mental illness [6]. Further consequences include a reduced quality of life, risks to physical safety, significant economic costs, and exclusion [6–8]. Therefore, investigating potentially effective therapeutic strategies for aggressive challenging behaviour in this population is paramount.

Existing research suggests interactions between biological, psychosocial and other environmental vulnerability factors in the presence and maintenance of aggressive challenging behaviour [9, 10]. Multiple factors (i.e. age, psychotropic medication use, pervasive developmental disorder, mood instability, etc.) are associated with an increased risk in adults with intellectual disability [11]. Additionally, significant heterogeneity within the intellectual disability population, and a person's cognitive and communication abilities, may have a significant impact on the phenomenology, severity, triggers and maintenance of aggressive challenging behaviour, as well as how it is understood by others [3, 12, 13]. Individuals with severe to profound intellectual disability may have more difficulties communicating their needs or thinking through the consequences of their actions compared to people with milder impairments. This heterogeneity needs to be considered when identifying and selecting appropriate interventions.

As aggressive challenging behaviour is a complex real-life problem with wide variation in population, presentation, causation, and context, a realist review can serve to contextualise the therapeutic impact of complex interventions, combining empirical and theoretical evidence to develop programme theories. Programme theories are based on the concept that underlying mechanisms (M) operate in particular contexts (C) to produce certain outcomes (O). Context-

mechanism-outcome (CMO) configurations are a means of producing programme theories, leading to a deeper understanding of how interventions work in diverse contexts and population groups [14–16]. Rapid realist reviews have been adapted from realist reviews to apply realist methodologies within shorter time constraints and involve expert knowledge users throughout the process [17].

This study aims to conduct a rapid realist review and develop a set of programme theories to explore how complex interventions work to reduce aggressive challenging behaviour, under which circumstances and for whom. Specifically, we will investigate:

1. Which interventions or intervention components work best to reduce aggressive challenging behaviour

2. Which contexts support or hinder their effectiveness

3. What are the key mechanisms that impact on the delivery, engagement and success of complex interventions

Where possible, we aim to identify key features of individuals with intellectual disability and of family and paid carers who respond differentially to complex interventions for aggressive challenging behaviour within care systems. In addressing these aims, we have integrated complementary approaches in our methodology: Identification of initial programme theories on what may sustain medium to long term change in treatment impact and practice; and a qualitative interview analysis to test these theories and factors associated with uptake and interventions delivery in routine care.

## Materials and methods

### Study design

Expert knowledge users contributed at each stage of the review process via a local reference group (LRG) and an expert panel. The LRG included nine stakeholders (i.e., practitioners, commissioners of services and family carers) recruited from charities and services in England, Scotland and Northern Ireland, who aimed to ensure the results were relevant to the clinical context of this population. The group met on three occasions between June 2020 and March 2021, and sent written feedback on the CMO configurations in March 2022. The expert panel included seven expert researchers from the study research team, aiming to ensure the review was focused and evidence was interpreted appropriately. They met on four occasions between July 2020-October 2021 and commented on the CMOs and if-then statements throughout this period.

We used a modified rapid realist review methodology [17] guided by the RAMESES-II standards for analysis and reporting [18], the process included the following stages and was guided throughout by the LRG and expert panel:

1. Development of the scope and initial programme theory

2. Literature searching, selection and appraisal of records (search terms are available in S1 Table)

3. Data extraction and analysis

4. Theory testing and validation via a qualitative interview analysis

5. Synthesis of findings

Further details of this process are outlined in Fig 1. Although presented sequentially, these stages were iterative and data extraction, analysis and programme theories were continually revised based on consultations with the LRG, expert panel, and interview findings. Our initial

---

**1. Development of the scope and initial programme theory**

| | |
|---|---|
| The core research team scoped the literature for:<br>- Qualitative and quantitative research<br>- Documents, operational guides and manuals describing intervention components, delivery and implementation<br>- Stakeholder experiences | Initial draft of the scoping document, relevant papers and initial CMO configurations presented to the LRG, expert panel and families/carers of individuals with intellectual disability. Feedback led to refined search terms and guided the development of the initial programme theory. |

**2. Literature searching, selection and appraisal of records**

Search terms were agreed with a librarian expert in systematic reviews. Six databases were searched on 27/07/2020 (PsycINFO, CINAHL, EMBASE, Medline, Health Management Information Consortium, Open Grey for grey literature). At each stage of the consultation process, sources were checked for additional papers and relevant articles were added to the review (n=12). Titles and abstracts were screened (5% independently screened by second reviewer). Full texts were then screened (20% independently screened). Disagreements were resolved through discussion and arbitration with a third reviewer. The quality of all studies were assessed.

**3. Data extraction and analysis**

Guided by the Template for Intervention Description and Replication checklist [19] and organised using Excel. This data was used to identify CMO configurations and develop explanatory theories in the form of 'if-then' statements. These were continually reviewed and revised.

**4. Theory testing and validation**

Theories were tested through six stakeholder interviews. A semi-structured interview guide was developed to explore experiences of interventions and whether these aligned with our emerging theories.

**5. Synthesis of findings**

An iterative process was followed to map evidence against theories identified from the literature to challenge or support them with programme theories tested by collating supporting evidence from the stakeholder interviews. This was then synthesised into a final programme theory.

**Fig 1. Stages of the rapid realist review [19].**

programme theory is presented in Fig 2. The theory outlines the context surrounding complex interventions for aggressive challenging behaviour, including system-level factors such as staffing, service model and therapist skills and person-level factors such as motivation, experience and ability. Intervention mechanisms were outlined as needing to focus on improving therapist self-efficacy, understanding of behaviour, encouraging reinforcement and providing adequate support to reduce aggressive challenging behaviour and other desired outcomes (e.g., increased carer efficacy, improved communication, etc.).

---

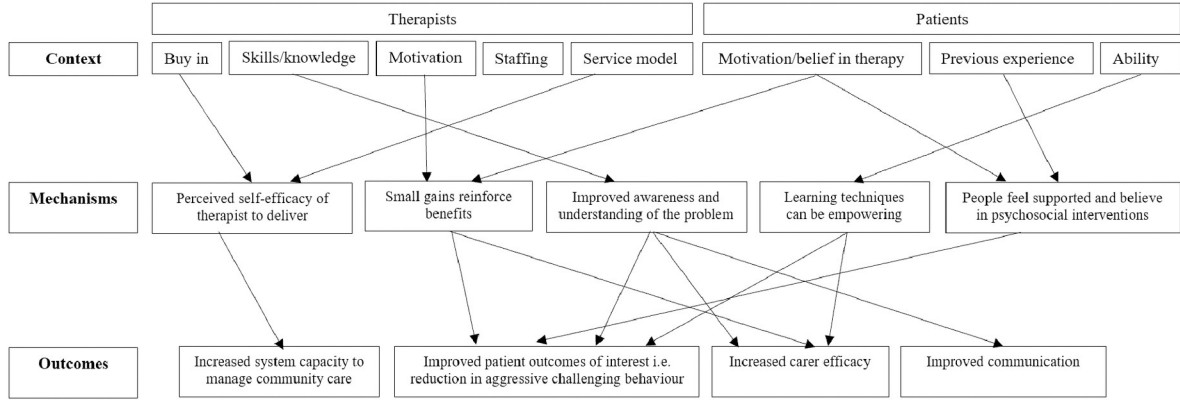

**Fig 2. Initial programme theory pathway for addressing aggressive challenging behaviour in adults with intellectual disability.**

Due to limited relevant data regarding addressing aggressive challenging behaviour in people with intellectual disability, findings from other population groups (e.g. older people with dementia and agitation, adults with mental illness displaying violence, children and young people with conduct problems) in a range of settings, including inpatient and forensic, provided useful information about the content and implementation of complex interventions in clinical pathways. Focusing on causal mechanisms and searching for additional sources that provide relevant information under different contexts is common in realist reviews [20]. As such, we retained broad inclusion criteria, looking beyond literature that sits fully within the field of intellectual disability and broadening the scope to explore all types of challenging behaviour.

Inclusion criteria were as follows:

1. Design: Qualitative, quantitative or mixed methods research, exploring interventions for challenging behaviour

2. Population: People demonstrating challenging behaviour with and without intellectual disability and/or autism, any type of mental illness, dementia, conduct or externalising disorders. The focus was on individuals aged over 18, although child/adolescent studies were also included if relevant to the research question.

3. Intervention: Programmes reporting outcome data for aggressive challenging behaviour or challenging behaviour

4. Outcomes of interest:

   - Individual outcomes–changes in challenging behaviour, incidents of behaviour and hospitalisations; service satisfaction, quality of life;

   - Family and paid carer outcomes–carer service satisfaction, quality of life, burden, competence to manage challenging behaviour;

   - Systems outcomes–staff knowledge/skills, staff engagement

## Quality appraisal

Quality was assessed according to relevance and rigour [18, 21].

A record was deemed more relevant if it met one or both criteria listed below:

A. Contributed significantly to theory building through its conceptual richness (the degree of theoretical and conceptual explanation of how an intervention is expected to work) [22]. We defined a conceptually rich record as one contributing to three or more initial programme theories, conceptualised as 'if/then statements.'

B. Recruited a sample of adults with intellectual disability in a community setting.

If a record met neither condition it was deemed less relevant. Judgements of relevance were made by means of an iterative and collaborative process of theory development with input from the expert panel.

Meanwhile, rigour refers to whether the methods employed by a study are credible [21]. Randomised control trials were deemed more rigorous if they received an overall score of 'low risk' or 'some concerns' in the Risk of Bias 2 measure, and less rigorous if they scored as 'high risk' [23]. Qualitative studies were judged as more rigorous if they received a total of 60% or above on the Critical Appraisal Skills Programme UK measure (scored from 0–100), or less rigorous if they did not [24]. All other study types were judged as more rigorous if they received a total score of 60% or above on the Mixed Methods Appraisal Tool (scored from 0–20), or less rigorous if they did not [25]. Two raters independently appraised the rigour of each record, with initial agreement at 74.6%. Any disagreements were then resolved through discussion. While judgements of relevance helped to ascertain the more important papers guiding theory development and rigour served to measure record quality, no records were excluded based on these judgements.

## Stakeholder interviews

Six stakeholders (4 healthcare professionals, 1 family carer, 1 service manager; 3 male) were recruited from England, Scotland and Northern Ireland and were interviewed between May-August 2021 about their experiences of receiving or delivering complex interventions for aggressive challenging behaviour. Informed written consent was obtained and stakeholders were interviewed through semi-structured interviews. These interviews were conducted to test and validate the emerging theories and included questions related to the nature, delivery and impact of interventions. Interviews were audio-recorded, transcribed and analysed thematically. The analysis was discussed with the LRG and expert panel to further refine the analysis and theories. Ethical approval was obtained to conduct this part of the review (REC reference: removed for blind submission).

## Patient and public involvement

In addition to working with expert knowledge users in the LRG throughout the review, two Patient and Public Involvement groups (one for family carers and one for service users) were consulted from the study design stage and throughout the review process during quarterly meetings. They reviewed the initial programme theories, CMO configurations, stakeholder interview schedule and final programme theories and their feedback was continually incorporated into revised iterations.

## Results

The initial searches identified 8343 records. After the removal of duplicates and the initial screening of abstracts and titles, 484 records underwent a full text search. 52 records were considered to fulfil the inclusion criteria. Following consultations with the LRG and expert panel up until March 2022, a further 12 relevant citation pearls were added (records identified that shared common characteristics with the other records under review) [26]. Five records were

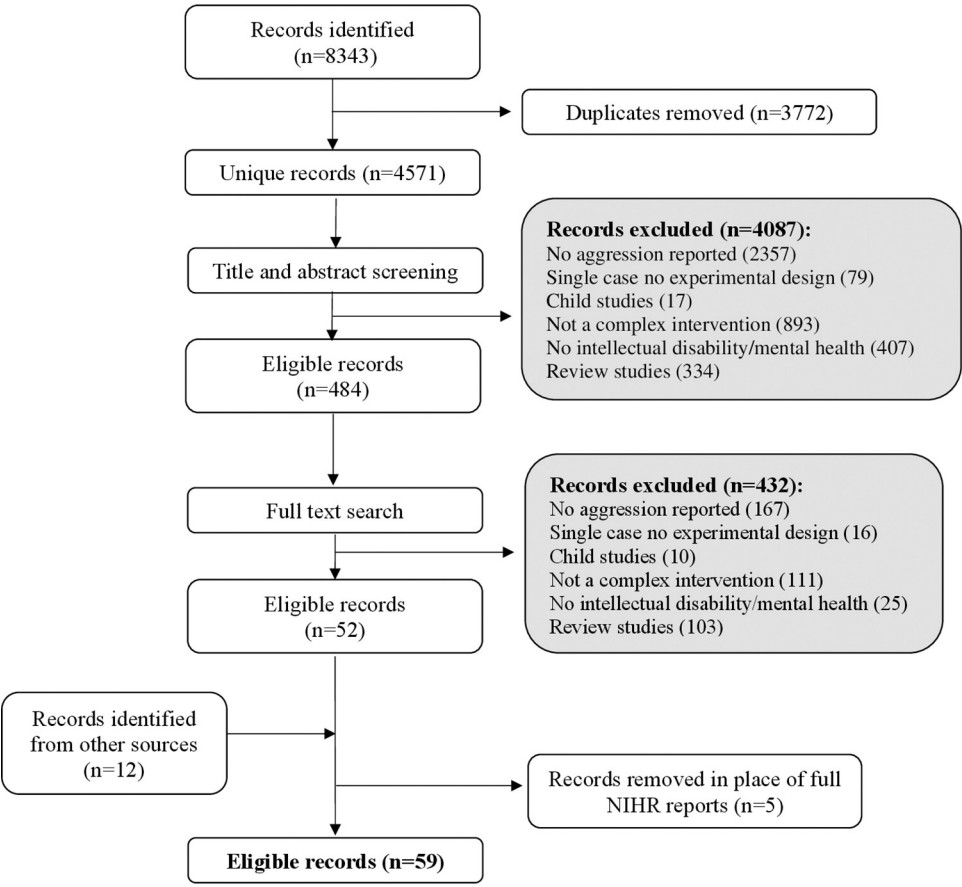

**Fig 3. PRISMA diagram.**

supplemented in place of full National Institute of Health Research (NIHR) reports which provided more detailed information. In total, 59 records were included in the review (see Fig 3 for full search details).

Of the 59 studies included, 37 focused on individuals with intellectual disability (mild-moderate n = 19, all levels of severity n = 11, moderate-severe or severe-profound n = 4, unspecified n = 3), 10 studies included neurotypical individuals with mental illness, 8 focused on people living with dementia, 2 included participants with behavioural disorders (i.e. patterns of disruptive behaviours that last for at least 6 months) and 2 on individuals with autism spectrum disorder. Six studies included just children or adolescents. Sample sizes ranged from 3 to 847 participants (mean: 99.82, SD = 172.17). Most studies were from the UK (n = 30), followed by the USA (n = 14), the Netherlands (n = 4), Canada (n = 3), Ireland (n = 2), New Zealand (n = 2) and one each in China, Australia, Sweden and Singapore.

Thirty studies used an experimental design (randomised controlled trial (n = 13), within-group repeated measures (n = 6), waiting list control (n = 5), multisite (n = 2), double cross-over (n = 1), non-randomised assignment to two intervention groups (n = 1), non-randomised assigned to matched control group (n = 1) and multiple baseline (n = 1)). The remaining 29 studies utilised the following designs: single case design (n = 8), feasibility/pilot (n = 6), qualitative (n = 5), protocols or intervention development/theoretical (n = 4), mixed methods (n = 3), observational (n = 2) and descriptive studies (n = 1).

Of the included studies, 30 reported on manualised interventions. Fifty reported single interventions, i.e., CBT approaches to anger management (n = 20), mindfulness-based approaches (n = 10), Positive Behavioural Support (n = 7), carer skills training (n = 7), Dialectical Behaviour Therapy (n = 4), multi-sensory intervention (n = 1) and an attachment-based therapeutic community intervention (n = 1). Nine papers reported on multi-component interventions, incorporating several approaches including behavioural activation, increasing meaningful events and promoting effective communication. Fourteen studies reported elements of personalisation, including tailoring the approach to the individual [27, 28], varying session length based on concentration levels [29] and creating individualised plans [30]. For a full summary of included studies, please see S2 Table.

In terms of quality, 47 papers were judged to be more relevant, with 34 papers contributing significantly to theory building and 26 papers recruiting a sample of adults with intellectual disability in the community. Thirteen papers satisfied both relevance and rigour and 12 papers met neither criterion. 43 papers were judged to be more rigorous. A total of 34 (57.6%) papers were judged to be both highly relevant and rigorous (see S3 Table).

We identified three overarching themes to capture key aspects of the evidence with sub-themes to account for the breadth of the data. The themes were 1. Working with the person displaying aggressive challenging behaviour, 2. Relationships and team focused approaches, and 3. Embedding and sustaining facilitating factors at team and systems levels. These theories are presented in Tables 1–3 below, including examples and supporting evidence from stakeholder interviews and our consultation work.

## 1. Working with the person displaying aggressive challenging behaviour

The following 4 CMO configurations were identified within this domain:

A. Emotion recognition, regulation and skill development

B. Approaches for individuals across all levels of intellectual disability

C. Meaningful activities

D. Facilitating factors of direct intervention

 i. Personalising intervention content, format and delivery

 ii. Intervention duration

**A. Emotion recognition, regulation and skill development**. A core element of many interventions related to supporting people to recognise and regulate their emotions, with a specific focus on managing feelings of anger. This requires individuals to recognise they are struggling to control their emotions and have the capacity and willingness to develop new functional skills and/or behaviours to manage those feelings. Only those with mild to moderate intellectual disability may have the sufficient cognitive skills required to benefit from this type of intervention.

Emotional regulation interventions have demonstrated efficacy and one participant with an intellectual disability stated the following after receiving cognitive behavioural therapy (CBT) for anger management: "I'm just the same person, but... if I get angry, I talk about what's annoying me... makes me feel much, what's the word, makes me feel much better... with myself" [31].

**B. Approaches for individuals across all levels of intellectual disability**. The population with intellectual disability is heterogenous, and those with more severe impairment may be unable to engage directly in the therapeutic process. Further, people with intellectual disability may have limited freedom and control over their lives, even if they have their own tenancies,

**Table 1. Working with the person displaying aggressive challenging behaviour.**

| Title | If (C) | Then (M) | Outcome (O) | Included studies that contribute to CMO | Supporting evidence from stakeholder interviews |
|---|---|---|---|---|---|
| **A. Emotion recognition, regulation, and skill development** | If individuals presenting with difficulties in emotional regulation have the necessary cognitive skills to engage in interventions delivered by lay therapists (family or paid carers) or clinically trained staff in both community and inpatient settings | they can learn: 1) To identify anger provoking situations/early signs of anger (triggers). 2) To distinguish between appropriate and inappropriate expression of anger. 3) To develop new, positive, skills or behaviours to replace less helpful ones. | This can lead to a reduction in the display of aggressive behaviour. | [29–31, 34–62] | *"People I have used it with have found that [cognitive behavioural therapy] really helps give them the tools for trying to modify their thinking in the future on their own if possible or even just with the help of staff scaffolding that. It almost sometimes feels like when they first get it, it is like a lightbulb moment for some people and when they first understand a link between thoughts and emotions." (Healthcare professional 1)* *"A massive bit of it is accessing the physiological component so that people are able to identify when they are very low down in their escalation cycle so that they are not waiting until they are really, really agitated before they do anything so that they can try and deescalate before it gets to that stage. A lot of people that I have worked with can't actually tell when they are getting angry and then don't know what to do about it at that stage. It is almost like it is a runaway train and they just can't stop." (Healthcare professional 1)* |
| **B. Approaches for individuals across all levels of intellectual disability** | If staff are trained to deliver sensory stimulation with service users in shared environments (including those with more severe/profound impairments and those requiring inpatient care), and/or staff are trained to provide comfort or create relaxing spaces | this can lead to a calmer environment, a reduction in sensory overload for individuals and an opportunity for de-escalation when anger-provoking situations occur. | This can lead to a reduction in the display of aggressive challenging behaviour and improved communication and relationships. | [28, 63–65] | *"Because an environment that is calming, that has greenery, that has colours on the walls, where there's not paint flaking off, where the echo is reduced, it's not over stimulating all the time. All of these things impact on how stressed we feel. And if we can reduce the stress in the environment, we stand a better chance of then actually helping someone to learn some new skills as well" (Healthcare professional 2)* *"To maybe looking at a range of sensory activities that will help them emotionally regulate, and that could be massage, music, it could be a focused activity." (Healthcare professional 3)* |

*(Continued)*

**Table 1.** (Continued)

| Title | If (C) | Then (M) | Outcome (O) | Included studies that contribute to CMO | Supporting evidence from stakeholder interviews |
|---|---|---|---|---|---|
| **C. Meaningful activities** | If staff are trained to organise and deliver meaningful activities (music, hobbies, tasks, social activities) with service users in shared environments (including those with more severe/ profound impairments and those requiring inpatient care) | unmet needs for sensory and social interaction can be met and personally meaningful activities can provide stimulation and combat boredom, whilst enabling service users to experience enjoyment from activities they are interested in. | This can lead to improvements in service user quality of life, while preventing incidences of aggressive behaviour and de-escalating incidences when they occur. | [66–69] | *"I had a guy in his 40s who was incredibly aggressive and a lot of attachment issues, and he discovered baking. I just wanted him to find one thing that he would be good at, and he loves baking, so he can actually make something. And the sense of achievement he gets from that and the sense of fulfilment he gets when he's able to give that to someone, and they go, that's actually really nice, and they mean it. And that's done more to manage his aggressive behaviours than anything they've done." (Healthcare professional 3)* |
| **D. Facilitating factors of direct intervention** <br> **i. Personalising intervention content, format and delivery** | If clinically trained therapists or adequately trained lay therapists (family or paid carers) personalise intervention content, session order, treatment pace and or duration and delivery format | interventions can address individuals' particular experiences, wishes, complex needs and abilities, and therefore can achieve a better fit. | This can then lead to greater engagement and satisfaction, all of which can then help to reduce the display of aggressive challenging behaviour. | [28, 29, 41–43, 48, 55, 56, 70] | *"Most stuff has been adapting existing programmes, or working on existing groups, that have been running, and thinking about the participants who are coming and how to make things accessible to them." (Healthcare professional 2)* <br> *"Also, length of sessions, it totally depends on how long somebody can concentrate for. . . "(Healthcare professional 1)* |
| **ii. Intervention duration** | If interventions are practiced by target individuals or their carers over longer treatment durations | individuals gain more opportunity to practice and embed skills. | Behaviour change is sustained, and aggressive behaviour is reduced. | [31, 44, 49, 61, 71–73] | *"We can keep going as long as it is necessary. It is really hard to say in terms of number of sessions." (Healthcare professional 1)* |

Note: C: Context, M: Mechanisms, O: Outcome

e.g., are placed in supported living, but with no control over who they live with or who supports them, and even less agency and control if living in residential or inpatient settings. A lack of autonomy over one's environments can be overstimulating, under-stimulating or stressful. Elements of sensory based interventions (e.g., music), can elicit positive emotions, whilst other sensory approaches (e.g., massage tools, lights, soft toys in a quiet and relaxing room, etc.) can be used to de-escalate situations by relaxing and distracting individuals, ideally in place of medication or other restrictive practices.

**C. Meaningful activities**. The opportunity for structured and personalised meaningful activities within both community and inpatient settings can combat boredom, provide stimulation, empower individuals and provide feelings of control over the environment. This can improve quality of life and can de-escalate or even prevent incidents of aggressive challenging behaviour.

**Table 2. Relationship and team focused approaches.**

| Title | If (C) | Then (M) | Outcome (O) | Included studies that contribute to CMO | Supporting evidence from stakeholder interviews |
|---|---|---|---|---|---|
| **A. Feeling listened to, valued, and supported in therapy** | If individuals with intellectual disability have access to a therapist who talks to them in a confidential, consistent, and non-judgemental way | they are more likely to feel respected and understood. | This can then increase engagement and facilitate positive treatment outcomes. | [56, 58, 61, 69, 71, 75] | *"It's that therapeutic relationship, nothing's going to work, no intervention is going to work unless the client has a really good relationship where they trust the person."* (Healthcare professional 3) *"Being open and honest with her. It is letting her know there are boundaries. . . I feel we have a positive relationship. And the respect, I do feel I try and get her to give me as much respect as what I will be giving her because at the end of the day, we're both human beings"* (Family carer) |
| **B. Supporting communication and relationships between service users and family carers** | If family members are taught to better communicate with and understand the person in their care and are taught to co-facilitate the delivery of interventions targeting aggressive challenging behaviour | this can lead to increased knowledge, empathy and understanding, a reduction in conflict or problematic interactions and improvements in the social environment of the family home. | This can lead to decreased aggressive behaviour and improved relationships at home. | [34, 70, 74, 76, 77] | *"I beat myself up at night when no-one's looking because I don't like [my child], or they annoy me, or I'm not meeting their needs and I'm failing them in every way. Or that I can't communicate [with] the professionals around me what they need and it seems to me that they're always being left to one side and never being appropriately cared for."* (Healthcare professional 4) *". . .and affording the listener, the mum, the ability to use that information to understand the situation then to respond in a way that then meets the need."* (Healthcare professional 4) |
| **C. Supporting communication and relationships between service users and paid carers** | If paid carers are taught to better communicate with and understand the person in their care and are taught to co-facilitate the delivery of interventions targeting aggressive challenging behaviour | this can help staff to have a better understanding of the behaviour and respond to individuals with greater levels of compassion and in a calm manner that facilitates de-escalation. | These processes can reduce rates of problematic interactions and build relationships as staff learn to adapt their responses, thereby reducing the display of aggressive challenging behaviour in service users. This can also lead to improved care practices, enhanced staff confidence and reduced burnout and stress. | [31, 50, 52, 60, 63, 65–69, 72, 73, 78–88] | *"If someone changes something in my schedule, I expect them to phone me and tell me why, and not apologise, but say sorry for the inconvenience. And the same with our clients, if you can't get the day-care and you're not going shopping when you usually do, when you don't have your one-to-one time, it shouldn't just be it's not happening. They need the respect of going, it's not happening because. . . And the language we use matters."* (Healthcare professional 3) *"We know our patients really, really well here because they are more longer-term patients, so we can see early warning signs that things might not be right. And then we can pick up then really quickly so they don't escalate up. And I think we're consistent."* (Service manager) |

*(Continued)*

**Table 2.** (Continued)

| Title | If (C) | Then (M) | Outcome (O) | Included studies that contribute to CMO | Supporting evidence from stakeholder interviews |
|---|---|---|---|---|---|
| **D. Facilitating factors of collaborative working between systems and families** | If the person with intellectual disability, their families, paid carers and professionals communicate efficiently without preconceived ideas or judgement and reflect on common goals and values | this can help build trusting relationships. | This can facilitate intervention adherence, buy in and better outcomes. | [30, 87] | *"We are all singing from the same hymn sheet. Not everybody is able to do the actual therapeutic part of it, but the whole team is in complete agreement with most of the strategies that get put in place because they just make common sense." (Healthcare professional 1)* *"Can email about it or chat about it and get together. I do feel. And if there was something wrong that I felt maybe that I wasn't equipped to deal with, I would know that I have them on support, on standby for the support should I need it." (Family carer)* |

Note: C: Context, M: Mechanisms, O: Outcome

**D. Facilitating factors of direct intervention**. *i. Personalising intervention content, format and delivery*. Personalisation in this context refers to improving the accessibility of an intervention through adaptations to promote understanding, engagement and to suit an individual's specific needs [32]. This can include:

- Adaptations–making adjustments (e.g., using pictures) to help individuals better understand content or by choosing to deliver elements based on their appropriateness and relevance to the individual (i.e., based on their communicative and cognitive abilities). Adaptations also apply to the pace and duration of sessions;

- Making intervention delivery fun and engaging can help with buy-in and motivation (e.g., sessions delivered through games to those with concentration difficulties).

Interventions personalised to address an individuals' experiences, wishes, needs and abilities can achieve a better fit and lead to more desired outcomes.

*ii. Intervention duration*. The duration of interventions ranged from a single session up to two years. Whilst it is still unclear whether longer durations are more efficacious than shorter ones, it is likely that a tiered approach is warranted depending on the complexity and severity of aggressive challenging behaviour [33]. For individuals with impaired cognitive ability, it is possible that longer treatment durations (32–52 weeks) can allow for maintenance which can help to embed skills and possibly support people to use them in real life situations. However, it was highlighted during consultations that family and paid carers may be unwilling or unmotivated to commit to longer term interventions, therefore shorter interventions may have greater uptake and adherence. See Table 1 for further information.

## 2. Relationships and team focused approaches

Four CMO configurations relate to the following:

A. Feeling listened to, valued, and supported in therapy

**Table 3. Sustaining and embedding change at team and systems levels.**

| Title | If (C) | Then (M) | Outcome (O) | Included studies that contribute to CMO | Supporting evidence from stakeholder interviews |
|---|---|---|---|---|---|
| **A. Removing barriers to mentorship and support for those delivering interventions** | If staff with varying levels of skills, abilities, and motivation receive regular supervision or mentorship from clinically trained staff, have intervention leads or champions and have choice in whether they are trained | individuals develop a clearer understanding of the intervention and what is expected from them as well as being motivated to embed these skills into routine practice, allowing for sustained and systemic change across numerous sites. | This can then help to ensure that interventions are delivered with high fidelity with treatment outcomes being maintained for longer with a reduction in aggressive behaviour observed. | [30, 31, 46, 50, 52, 59, 63, 65–67, 77, 84–88] | *"Staff have said when they've been going back to their areas, how [support] made them feel really relieved and comforted. And that if they were worried about anything, then they could come and speak to us. As well as the staff team being really welcoming, it's had the positive impact that's gone back to them."* (Service manager) |
| **B. Intervention deliverers having protected time to learn and practice skills** | If carers and staff facilitating intervention delivery to people with intellectual disability have protected time to practice and learn new skills | this can promote engagement and ensure they embed new skills into practice. | This can then increase a sense of competence and confidence in applying new skills and help to ensure changes in behaviour are sustained. | [34, 51, 54, 70, 74, 76, 78, 83] | *"...allows staff... to air their views on situations that have happened, review how a situation was dealt with and managed. Come up with ideas and formulations on how to manage situations in a better way next time, they're able to throw all their ideas out..."* (Service manager) |
| **C. Facilitating factors for collaborative working within teams** | If paid carers and staff are taught interventions which focus on working cohesively and sharing responsibilities with other staff members | interventions will run smoothly, and staff can share skills and support others to deliver the intervention whilst also building a shared understanding of the triggers and maintenance of behaviour. | This can result in more positive shared environments, trust and collective responsibility, improving staff and service user outcomes (e.g., sustained reductions in aggressive challenging behaviour, decreased staff turnover, burnout and stress) | [31, 63, 65, 79, 81, 85, 87] | *"...it supports the staff to be consistent and come up with consistent approaches to help support them in managing the individual, and also give the individual consistency".* (Service manager) |

Note: C: Context, M: Mechanisms, O: Outcome

B. Supporting communication and relationships between service users and family carers

C. Supporting communication and relationships between service users and paid carers

D. Facilitating factors of collaborative working between systems and families

Whilst there is overlap between B) and C) CMO configurations above, they address the theme from different perspectives and highlight distinct elements to be considered based on the type of carer.

**A. Feeling listened to, valued, and supported in therapy.** Individuals value the opportunity to speak to a receptive therapist who 'understands how they feel,' treats them respectfully and speaks to them in a confidential, consistent, and non-judgemental way. Therapists can become allies or attachment figures, and therapeutic relationships characterised by warmth and empathy encourage individuals to learn new ways to manage their emotions and recognise and respond adaptively to situations. An individual with intellectual disability receiving CBT reported feeling respected by their therapist: "Well [my therapist] seems to think I've got the brain of an adult, she seems to think I speak like an adult and I do things in an adult way" [71].

**B. Supporting communication and relationships between service users and family carers**. Various interventions focused on improving and enhancing relationships between service users and family carers. Family members were taught to:

1. Care for themselves in more mindful ways (to reduce stress), whilst having their feelings validated through support and reassurance from therapists;

2. Better understand the causes, triggers and what maintains aggressive challenging behaviour and develop skills to increase self-efficacy to respond to behaviour more adaptively;

3. Respond positively to incidences of aggressive challenging behaviour with greater empathy and acceptance.

This can help to increase carer empathy, improve the home social environment and reduce conflict, carer stress and aggressive challenging behaviour. Our LRG noted that families may be overwhelmed by basic unmet needs and may have diminished emotional and/or physical resources to learn and implement such changes. However, for those that do have the time and resources, these changes can have a significant and broader impact: "Part of the transformation. . .appears to be changes in the way they [family carers] relate to all events in their environment, rather than the acquisition of a set of skills to specifically change their children's behaviors."–(Mindful parenting intervention [74]).

**C. Supporting communication and relationships between service users and paid carers**. Interventions focusing on training paid carers and staff in inpatient units, residential homes or from community services, taught staff to:

1. Get to know the individual, viewing them as a person, not a patient;

2. Learn skills and feel confident to use de-escalation when necessary to reduce arousal;

3. Reduce incidences of conflict and improve environmental conditions (e.g. mitigating bad news, increasing socially meaningful activities).

Our LRG added that building relationships can lead to increased compassion by paid carers and a better understanding of behaviour. This can improve carer confidence and wellbeing, positively impacting care practices. Paid carers can learn to adapt their responses to the context and respond in a way that de-escalates the situation to reduce occurrences of aggressive challenging behaviour.

**D. Facilitating factors of collaborative working between systems and families**. Service users, their families, paid carers and professionals can build functional and collaborative relationships with one another by:

1. Communicating efficiently, with carers/professionals providing families with sufficient information about the person's care, reflecting on common values/goals and shared responsibilities;

2. Carers/professionals remembering that love underpins a family's motivation for seeking sufficient support and potential frustrations when services do not meet expectations, rather than judging family members as demanding or hard to reach.

This results in more trusting relationships and a collaborative effort to achieve the best possible outcomes for the individual. See Table 2 for details of CMO configurations for this domain.

## 3. Sustaining and embedding change at team and systems levels

This theory encompasses three domains presented in Table 3. These are the following:

A. Removing barriers to mentorship and support for those delivering interventions

B. Intervention deliverers having protected time to learn and practice skills

C. Facilitating factors for collaborative working within system teams

**A. Removing barriers to mentorship and support for those delivering interventions**. Intervention delivery in pragmatic conditions often means some elements may not be delivered as intended or other factors may impact the outcome. Therapists are likely to have varying levels of knowledge, skills, motivation and abilities, and require support to exercise autonomy and choice around whether and how they are involved in intervention delivery. Capability can be enhanced by:

1. Receiving regular support, training and/or mentorship from clinicians and/or qualified trainers. This can support with motivating therapists, as well as providing them with opportunities to engage in reflective learning;

2. Managers working within services or residential care supporting strategies to enhance implementation, such as nominating intervention leads/champions to support, mentor and motivate staff. This can encourage learning, enhance fidelity, and promote the enhancement of staff skills, abilities, and confidence;

3. Training motivated staff (within services or residential care) as lay therapists who have an interest in delivering the intervention. This can incentivise, engage and enthuse staff to deliver the therapies effectively.

If staff have limited time and resources to dedicate to the extra responsibilities associated with delivering new interventions and do not receive regular support, these interventions will not be conducted with good fidelity and may not be delivered consistently, meaning reductions in aggressive challenging behaviour will not be observed.

**B. Intervention deliverers having protected time to learn and practice skills**. If family or paid carers are facilitating intervention delivery, time should be allocated for them to practice and learn new skills (e.g., at convenient times within the family home, protected time in wards or supported living environments). Carers will then feel valued and prioritised, and practice will boost their confidence to embed these skills within daily routines.

**C. Facilitating factors for collaborative working within teams**. Staff working cohesively and across boundaries to share responsibilities (e.g., through regular meetings where teams share perspectives and plan goals) can help interventions run with good fidelity. Staff build a shared understanding of the nature of aggressive challenging behaviour and can pass on skills and reflections upon what works for specific people to other staff, facilitating intervention uptake into routine practice and allowing for organisational change through a shared reflective process and collective responsibility. This in turn can also result in more positive shared environments which can improve both staff and service user outcomes, such as sustained reductions in aggressive behaviour, decreased staff burnout and improved service user and staff quality of life. See Table 3 for further details.

## Final programme pathway

The full final programme pathway incorporating all programme theories for addressing aggressive challenging behaviour in adults with intellectual disability is presented in Fig 4. The context includes person-level factors related to specific approaches for addressing aggressive challenging behaviour and facilitating factors that enhance their effectiveness (e.g., personalisation based on needs and abilities); relational factors between the person with learning

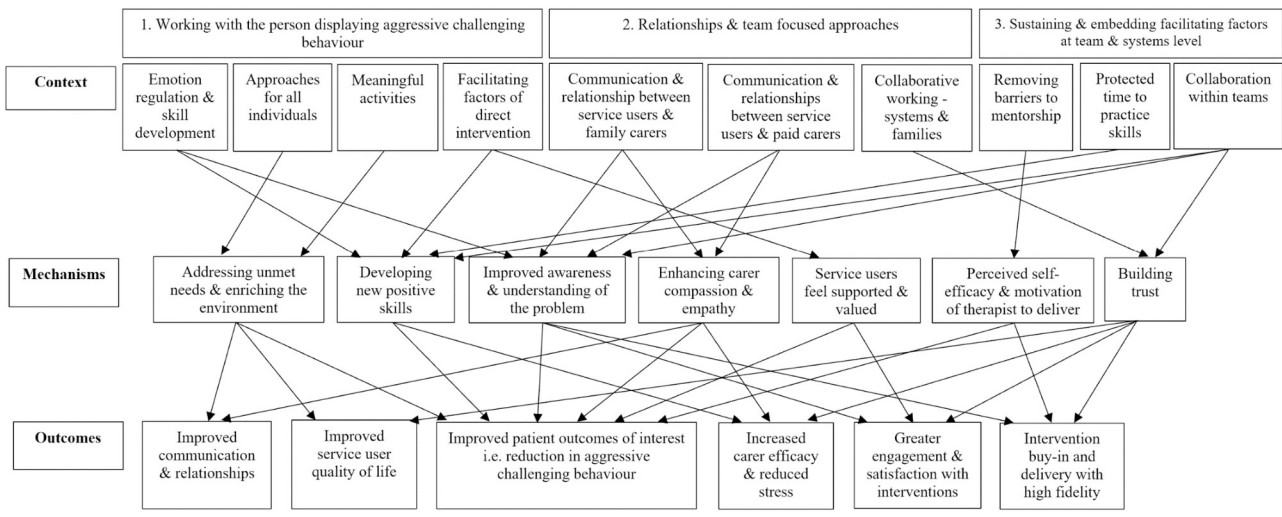

**Fig 4. Final programme theory pathway for addressing aggressive challenging behaviour in adults with intellectual disability.**

disability, their family and paid carers and professionals; system level factors related to collaboration, mentorship and protected time. Intervention mechanisms should focus on skill building, addressing unmet need, enriching the environment, enhancing understanding of aggressive challenging behaviour, improving carer compassion, enhancing therapist self-efficacy and motivation, supporting the service user, and building trust. If these are addressed, this should facilitate the desired outcomes of improved communication and relationships, improved quality of life, increased carer efficacy, greater buy-in, engagement and satisfaction with interventions and a reduction in aggressive challenging behaviour.

## Discussion

### Key findings

This rapid realist review aimed to explore the mechanisms behind complex interventions addressing aggressive challenging behaviour, elucidating how they work in practice and for whom by developing programme theories through contexts-mechanism-outcome configurations. The review included 59 studies. We identified 11 CMOs within three domains to understand how complex interventions work in practice for adults with intellectual disability who display aggressive challenging behaviour: working with the person displaying aggressive challenging behaviour, relationships, and team focused approaches, and sustaining and embedding facilitating factors at team and systems levels.

We identified emotional regulation training, sensory based approaches, and the inclusion of meaningful activities as key components of complex interventions that can effectively support people to reduce aggressive challenging behaviour, improve relationships and the person's quality of life. These approaches work through enriching the environment, addressing unmet needs and through facilitating the development of positive skills. However, approaches that require more cognitive and communicative abilities (i.e. learning skills to control and manage emotions or directly learning mindfulness techniques) may only be appropriate for the subset of the population with milder intellectual impairment [40, 89]. Many available interventions are administered to people with intellectual disability regardless of severity and this may be over-inclusive and ineffective, as some people may not have the capacity to benefit from the chosen approach. Hence, interventions should be specifically chosen to suit an individual's

ability level and a suitable intervention duration needs to also be considered. Positive outcomes can be further facilitated when elements of chosen interventions are further personalised and tailored to the person and when there is an opportunity for individuals, carers and staff to practice and embed the skills they learn.

The majority of behavioural change interventions in the general population focus on the individual [90, 91], however it is evident from our review that carer involvement and collaborative relationships are crucial to facilitate and sustain change in cognitively impaired populations. Family and paid carers are often on the receiving end of aggressive challenging behaviour, which affects their relationship with the person, as well as how they interact and respond when incidents occur. Carers often also experience burnout and may lack motivation and confidence [92, 93], therefore interventions that address these barriers (e.g. through improving awareness and understanding, enhancing carer compassion and empathy) are equally essential to reduce carer stress, increase efficacy, build trust and improve outcomes.

To enhance the acceptability of interventions, there also needs to be an additional focus on effective delivery and implementation, through providing adequate training, protected time to practice skills, collaborative working within teams, and through continual mentorship and support to staff [94]. The quality of staff training in an intervention may be more influential in supporting the achievement of desired outcomes than the content or characteristics of the intervention itself [95]. Thus, it is essential for there to be buy-in at senior management level to ensure appropriate training is delivered and to provide cohesion, clear leadership and a supportive and motivating environment. This will promote therapist self-efficacy and motivation to deliver the intervention, leading to higher intervention fidelity and greater engagement and satisfaction from recipients.

## Strengths and limitations

To our knowledge, this is the first rapid realist review for complex interventions addressing aggressive challenging behaviour. Therefore, this review provides novel insights that are likely to be absent in the current literature. We used a rigorous analysis process including a comprehensive literature search, the inclusion of grey literature, consultations with stakeholders and expert researchers, and input from stakeholder interviews to ensure the work captured the perspectives of those with lived experience. We believe that including evidence from other populations with relevant characteristics addressed an important gap, as many adapted or modified complex interventions for people with intellectual disability have been examined in small feasibility or pilot studies, and therefore may lack methodological power and robustness. Overall, the included studies were determined to be of good quality and the majority contributed significantly to the building and development of theories. Over half of the studies were also specifically relevant to the intellectual disability population.

However, despite using an iterative search strategy, some relevant studies may have been missed, although consultations with academic experts and the inclusion of citation pearls should have reduced this likelihood. We were only able to obtain six interviews for the review referring to a limited set of interventions, e.g., CBT informed anger management, Dialectical Behaviour Therapy and Positive Behaviour Support, due to the complexities of the Covid-19 pandemic and this was fewer than intended. The studies included do not address ethnic diversity and the cultural appropriateness of interventions, and there may be additional challenges and adaptations that need considering for these families. Finally, all included studies were conducted prior to the pandemic and there may be additional implications for the delivery and effectiveness of complex interventions with the increasing use of tele-mental health that warrant further investigation.

### Implications for clinical practice

This review highlights the importance of understanding the needs of the individual and the circumstances and context surrounding behavioural presentations to provide personalised and targeted support. It is evident that a one-size-fits-all generalised approach to address aggressive challenging behaviour is inappropriate for this population, and it is essential to understand and review a person's capabilities to choose an intervention they will be able to engage with and benefit from.

System adoption and the provision of appropriate staff support ensures optimal conditions for effective intervention delivery and by ensuring therapists are motivated and committed. This in turn promotes higher intervention fidelity, improved patient outcomes and greater patient engagement and satisfaction with services.

### Implications for policy

There is wide variability in available service provision and care for people with intellectual disability who display aggressive challenging behaviour. Pharmacological interventions are frequently used, despite a paucity of robust evidence for their efficacy and with the risk of significant side-effects [6, 96]. Whilst current policy emphasises the importance of personalisation in psychosocial interventions, policy makers, and those in positions of commissioning services and launching national initiatives, must ensure there is sufficient investment in skilled staff, training and resources that allow for the delivery and implementation of personalised therapies. These therapies also need to combat the attitudes and beliefs of those supporting the programme and include the explicit use of behavioural change techniques. Associated work should address health disparities, social connectedness, previous trauma and other influences at a familial and individual level.

### Future directions

Further work is needed to explore integrated, personalised and targeted approaches to address aggressive challenging behaviour, whilst also utilising robust study designs [6] and to investigate specific pathogenetic mechanisms cross-sectionally and across time. Future research should also consider and focus on ethnically diverse groups and intervention implementation (i.e., to explore barriers related to staffing and funding) within services.

## Conclusion

Aggressive challenging behaviour is a primary driver for hospital admissions and the use of restrictive practices in individuals with intellectual disability. This results in high individual and economic costs and highlights the importance of identifying effective treatment approaches. Complex interventions can be efficacious to address aggressive challenging behaviour in individuals with intellectual disability and in other populations, although they need to be personalised and should potentially address several problems in parallel. The inclusion of family and paid carers within interventions is essential to facilitate improved communication, relationships, the embedding of skills and the reduction of aggressive challenging behaviour. Further work is needed to ensure the effective implementation of these interventions through services.

## Supporting information

**S1 Checklist. RAMESES-II checklist.**
(DOCX)

**S1 Table. Search strategy for MEDLINE, EMBASE, PsycINFO and HMIC using the Ovid interface on 27-07-20.**
(DOCX)

**S2 Table. Summary of studies included in the rapid realist review.**
(DOCX)

**S3 Table. Relevance and rigour judgements as a means of quality appraisal.**
(DOCX)

**S1 File. Prospero registration.**
(PDF)

**S1 Data. Extracted data.**
(XLSX)

## Acknowledgments

We would like to extend a special thanks to all the contributors and members of our expert panel, LRG and study team. We would also like to thank the members of our patient advisory groups who contributed to this review. We thank Fernanda Fenn Torrente (trained medical student) for her contributions to appraising the rigour of each record for the quality assessment.

## Author Contributions

**Conceptualization:** Angela Hassiotis, Andrew Jahoda, Afia Ali, Umesh Chauhan, Sally-Ann Cooper, Liz Steed, Andre Strydom, Laurence Taggart, Penny Rapaport.

**Data curation:** Penny Rapaport.

**Formal analysis:** Stephen Naughton, Angela Hassiotis, Penny Rapaport.

**Funding acquisition:** Angela Hassiotis, Andrew Jahoda, Afia Ali, Umesh Chauhan, Sally-Ann Cooper, Liz Steed, Andre Strydom, Laurence Taggart, Penny Rapaport.

**Investigation:** Stephen Naughton.

**Methodology:** Angela Hassiotis, Andrew Jahoda, Afia Ali, Umesh Chauhan, Sally-Ann Cooper, Liz Steed, Andre Strydom, Laurence Taggart, Penny Rapaport.

**Project administration:** Angela Hassiotis, Afia Ali, Penny Rapaport.

**Supervision:** Angela Hassiotis, Penny Rapaport.

**Validation:** Angela Hassiotis, Penny Rapaport.

**Visualization:** Rachel Royston, Stephen Naughton.

**Writing – original draft:** Rachel Royston, Stephen Naughton, Penny Rapaport.

**Writing – review & editing:** Rachel Royston, Angela Hassiotis, Andrew Jahoda, Afia Ali, Umesh Chauhan, Sally-Ann Cooper, Athanasia Kouroupa, Liz Steed, Andre Strydom, Laurence Taggart, Penny Rapaport.

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
