## [Decision Letter · Decision Letter 0]

4 Apr 2023

PONE-D-23-05137Complex interventions for aggressive challenging behaviour in adults with intellectual disability: a rapid realist review informed by multiple populationsPLOS ONE

Dear Dr. Royston,

Thank you for submitting your manuscript to PLOS ONE. Two reviewers have evaluated your manuscript and minor revisions are recommended. Therefore, I invite you to submit a revised version of the manuscript that addresses the points raised during the review process.

We look forward to receiving your revised manuscript.

Kind regards,

Weifeng Han, PhD

Academic Editor

PLOS ONE

Journal Requirements:

Reviewers' comments:

Reviewer's Responses to Questions

**Comments to the Author**

1. Is the manuscript technically sound, and do the data support the conclusions?

Reviewer #1: Yes

Reviewer #2: Yes

2. Has the statistical analysis been performed appropriately and rigorously? 

Reviewer #1: Yes

Reviewer #2: Yes

3. Have the authors made all data underlying the findings in their manuscript fully available?

Reviewer #1: Yes

Reviewer #2: Yes

4. Is the manuscript presented in an intelligible fashion and written in standard English?

Reviewer #1: Yes

Reviewer #2: Yes

5. Review Comments to the Author

Reviewer #1: The study aimed to explore the mechanisms that contribute to the successful implementation of interventions for aggressive challenging behavior in individuals with intellectual disability. The review included 59 studies and developed three domains and 11 context-mechanism-outcome configurations. Overall, the study highlights the importance of person-centered approaches, team-focused approaches, and sustaining and embedding facilitating factors at team and systems levels.

The methodology used for this review is reasonable and comprehensive. The study offers novel insights into the mechanisms that contribute to successful interventions for aggressive challenging behaviors in individuals with intellectual disability. It is also noted that the study limitations have been appropriately listed.

Reviewer #2: It is so interesting and well written article. But some minor comments could be addressed.

1- The discussion is well written and organized but the authors did not discuss regarding their results about main findings.

2-It is better the aims and research questions wrote more clearly.

6. PLOS authors have the option to publish the peer review history of their article (what does this mean?). If published, this will include your full peer review and any attached files.

Reviewer #1: No

Reviewer #2: **Yes: **Marsa Gholamzadeh

---

## [Author Response · Author response to Decision Letter 0]

25 Apr 2023

Dear Editors,

Thank you for your consideration and comments on our manuscript “Complex interventions for aggressive challenging behaviour in adults with intellectual disability: a rapid realist review informed by multiple populations” (PONE-D-23-05137).

We would like to thank both reviewers for their comments and suggestions on the manuscript. We have addressed the following points:

Reviewer #1:

The study aimed to explore the mechanisms that contribute to the successful implementation of interventions for aggressive challenging behavior in individuals with intellectual disability. The review included 59 studies and developed three domains and 11 context-mechanism-outcome configurations. Overall, the study highlights the importance of person-centered approaches, team-focused approaches, and sustaining and embedding facilitating factors at team and systems levels.

The methodology used for this review is reasonable and comprehensive. The study offers novel insights into the mechanisms that contribute to successful interventions for aggressive challenging behaviors in individuals with intellectual disability. It is also noted that the study limitations have been appropriately listed.

Response:

We would like to thank this reviewer for their positive review of our manuscript, and we are pleased the reviewer agrees that this study offers novel insights in this topic area.

Reviewer #2:

 It is so interesting and well written article. But some minor comments could be addressed.

1- The discussion is well written and organized but the authors did not discuss regarding their results about main findings.

2-It is better the aims and research questions wrote more clearly.

Response:

1 – The ‘key findings’ section in the discussion outlines the main results of the study. We have edited this section to map the content of this section more closely to the final programme pathway figure (Figure 4). We hope this clarifies the main findings of the study in line with this suggestion.

The changes to the text have been highlighted below in bold.

‘We identified emotional regulation training, sensory based approaches, and the inclusion of meaningful activities as key components of complex interventions that can effectively support people to reduce aggressive challenging behaviour, improve relationships and the person’s quality of life. These approaches work through enriching the environment, addressing unmet needs and through facilitating the development of positive skills. However, approaches that require more cognitive and communicative abilities (i.e. learning skills to control and manage emotions or directly learning mindfulness techniques) may only be appropriate for the subset of the population with milder intellectual impairment [39, 88]. Many available interventions are administered to people with intellectual disability regardless of severity and this may be over-inclusive and ineffective, as some people may not have the capacity to benefit from the chosen approach. Hence, interventions should be specifically chosen to suit an individual’s ability level and a suitable intervention duration needs to also be considered. Positive outcomes can be further facilitated when elements of chosen interventions are further personalised and tailored to the person and when there is an opportunity for individuals, carers and staff to practice and embed the skills they learn. 

The majority of behavioural change interventions in the general population focus on the individual [89, 90], however it is evident from our review that carer involvement and collaborative relationships are crucial to facilitate and sustain change in cognitively impaired populations. Family and paid carers are often on the receiving end of aggressive challenging behaviour, which affects their relationship with the person, as well as how they interact and respond when incidents occur. Carers often also experience burnout and may lack motivation and confidence [91, 92], therefore interventions that address these barriers (e.g. through improving awareness and understanding, enhancing carer compassion and empathy) are equally essential to reduce carer stress, increase efficacy, build trust and improve outcomes.

To enhance the acceptability of interventions, there also needs to be an additional focus on effective delivery and implementation, through providing adequate training, protected time to practice skills, collaborative working within teams, and through continual mentorship and support to staff [93]. The quality of staff training in an intervention may be more influential in supporting the achievement of desired outcomes than the content or characteristics of the intervention itself [94]. Thus, it is essential for there to be buy-in at senior management level to ensure appropriate training is delivered and to provide cohesion, clear leadership and a supportive and motivating environment. This will promote therapist self-efficacy and motivation to deliver the intervention, leading to higher intervention fidelity and greater engagement and satisfaction from recipients.’

(Pages 26-27, lines 470-505).

2 – We agree that the research aims and questions could be stated more explicitly. We have revised the end of the introduction as follows:

‘This study aims to conduct a rapid realist review and develop a set of programme theories to explore how complex interventions work to reduce aggressive challenging behaviour, under which circumstances and for whom. Specifically, we will investigate:

1) Which interventions or intervention components work best to reduce aggressive challenging behaviour

2) Which contexts support or hinder their effectiveness 

3) What are the key mechanisms that impact on the delivery, engagement and success of complex interventions

Where possible, we aim to identify key features of individuals with intellectual disability and of family and paid carers who respond differentially to complex interventions for aggressive challenging behaviour within care systems. In addressing these aims, we have integrated complementary approaches in our methodology: Identification of initial programme theories on what may sustain medium to long term change in treatment impact and practice; and a qualitative interview analysis to test these theories and factors associated with uptake and interventions delivery in routine care.’ 

(Page 4, lines 76-90) 

We hope these changes are considered as satisfactory by the reviewers. Please let us know if you require any additional information. 

Kind regards,

Rachel Royston, Stephen Naughton, Angela Hassiotis, Andrew Jahoda, Afia Ali, Umesh Chauhan, Sally-Ann Cooper, Athanasia Kouroupa, Liz Steed, Andre Strydom, Laurence Taggart & Penny Rapaport.

---

## [Editor Report · Decision Letter 1]

27 Apr 2023

Complex interventions for aggressive challenging behaviour in adults with intellectual disability: a rapid realist review informed by multiple populations

PONE-D-23-05137R1

Dear Dr. Royston,

We’re pleased to inform you that your manuscript has been judged scientifically suitable for publication and will be formally accepted for publication once it meets all outstanding technical requirements.

Kind regards,

Weifeng Han, PhD

Academic Editor

PLOS ONE
---

## [Editor Report · Acceptance letter]

10 May 2023

PONE-D-23-05137R1 

Complex interventions for aggressive challenging behaviour in adults with intellectual disability: a rapid realist review informed by multiple populations 

Dear Dr. Royston:

I'm pleased to inform you that your manuscript has been deemed suitable for publication in PLOS ONE. Congratulations! Your manuscript is now with our production department. 

Kind regards, 

on behalf of

Dr. Weifeng Han 

Academic Editor

PLOS ONE